# A generative adversarial network model alternative to animal studies for clinical pathology assessment

Xi Chen [1], Ruth Roberts[2,3], Zhichao Liu [1,4] ✉ & Weida Tong [1] ✉

Animal studies are unavoidable in evaluating chemical and drug safety. Generative Adversarial Networks (GANs) can generate synthetic animal data by learning from the legacy animal study results, thus may serve as an alternative approach to assess untested chemicals. AnimalGAN, a GAN method to simulate 38 rat clinical pathology measures, was developed with significant robustness even for the drugs that vary significantly from these used during training, both in terms of chemical structure, drug class, and the year of FDA approval. AnimalGAN showed comparable results in hepatotoxicity assessment as using the real animal data and outperformed 12 conventional quantitative structure-activity relationship approaches. Using AnimalGAN, a virtual experiment of 100,000 rats ranked hepatotoxicity of three structurally similar drugs in a similar trend that has been observed in human population. AnimalGAN represented a significant step with artificial intelligence towards the global effort in replacement, reduction, and refinement (3Rs) of animal use.

Animal studies are pivotal in biomedical sciences for understanding disease progression, discovering prognosis/diagnosis biomarkers, risk and safety assessments, and developing novel treatment options[1]. Since animal studies generate multidimensional information, much of which parallels metrics taken in the clinical setting, animal data are critical in characterizing the risk and safety profiles of many chemical and drug products under the jurisdiction of the Food and Drug Administration (FDA). Nonetheless, focus has shifted from traditional animal work for assessing human safety[2,3]. The FDA Modernization Act 2.0 has recently been signed into law by the President, which emphasizes a need to explore alternative options that support the 3Rs (Replacement, Reduction, and Refinement)[4] of animal use.

Artificial Intelligence (AI) technology has driven innovation in many spheres, including toxicology, where AI has the potential to form alternative approaches for risk assessment in support of the 3Rs[5,6]. Most AI application in toxicology has been largely used for analyzing and processing data to identify patterns and make predictions such as Quantitative Structure Activity Relationships (QSARs). Often, these approaches are focused on predicting a single endpoint representing a high-level abstraction (e.g., toxic vs. non-toxic) of toxicological effects[7]. In contrast, animal-based toxicity assessments provide much richer, multi-dimensional information to support risk assessment and decision making, such as that derived from toxicogenomics and clinical pathology.

Recently, generative AI has gained tremendous momentum such as ChatGPT. Generative AI such as Generative Adversarial Networks (GANs)[8] can create new content, thus lending itself as an alternative approach to generate synthetic animal data of untested chemicals by learning from the legacy animal study results. We reported here a generative AI model for animal studies, called AnimalGAN, which was developed using a GAN method. AnimalGAN were able to simulate a virtual animal experiment to generate multi-dimensional profiles similar to those obtained from traditional animal studies. Specifically, AnimalGAN established the association between chemical exposure (the combination of chemical, dose, and exposure duration) and clinical pathology findings (e.g., clinical chemistry and hematologic measures) in legacy animal study data to generate synthetic clinical

[1]National Center for Toxicological Research, Food and Drug Administration, Jefferson, AR 72079, USA. [2]ApconiX Ltd, Alderley Park, Alderley Edge SK10 4TG, UK. [3]University of Birmingham, Edgbaston, Birmingham B15 2TT, UK. [4]Currently working at Integrative Toxicology, Nonclinical Drug Safety, Boehringer Ingelheim Pharmaceuticals, Inc., Ridgefield, CT 06877, USA. ✉e-mail: zhichao.liu@boehringer-ingelheim.com; Weida.Tong@fda.hhs.gov

pathology profiles for new and untested drugs and other chemicals in a predefined dose and treatment duration.

We demonstrated that the AnimalGAN approach was robust even for the drugs that vary significantly from these used during training, both in terms of chemical structure, drug class, and the year of FDA approval. By comparing to the conventional computational toxicology methods, AnimalGAN outperformed the 12 traditional quantitative structure-activity relationship (QSARs) methods in predicting all clinical pathology measures. Moreover, AnimalGAN results were comparable to animal studies in assessing hepatotoxicity of drugs. One of the most critical arguments in 3Rs science is that animal studies do not always predict human outcomes in complex conditions such as idiosyncratic drug induced liver injury (iDILI)[2]. We found that, however, AnimalGAN could approximate populations of diverse individual animal clinical pathology data by conducting unlimited synthetic experiments (as the resource allows), which offers an opportunity to detect rare toxicological events that almost certainly would not be possible to identify in traditional animal studies, thus improving the translation of animal data to human outcomes.

## Results

### AnimalGAN development

In the present study, AnimalGAN generated 38 clinical pathology measurements, where the test compounds are represented by chemical descriptors along with the study conditions of treatment duration (3, 7, 14, and 28-day) and dose groups (high, medium, and low). As depicted in Fig. 1, the AnimalGAN model was developed on 6442 rats (the training set) corresponding to 110 compounds (most are drugs) under 1317 treatment conditions (a combination of compound-dose-time) from the Open Toxicogenomics Project-Genomics Assisted Toxicity Evaluation Systems (TG-GATEs) database[9] with a hybrid GANs architecture (see Methods and Fig. 1a). The model was then evaluated with 1636 rats (the test set) under 332 treatment conditions for 28 different compounds from the same TG-GATEs database (Fig. 1b). The high concordance between the synthetic and real clinical pathology measurements in the test set was observed with low root-mean-square error (RMSE) (17.58, which is significantly smaller than the median of background control 72.46 with a Wilcoxon rank-sum test $p$ value $2.48 \times 10^{-169}$, Fig. 1c) and high cosine similarity (1.00, which is significantly higher than the median of background control 0.98 with a Wilcoxon rank-sum test $p$ value $1.45 \times 10^{-181}$, Fig. 1d). Visualization using $t$-distributed stochastic neighbor embedding ($t$-SNE) dimensionality reduction also showed the high similarity between the synthetic and real clinical pathology measurements for the test set (Fig. 1e). Supplementary Fig. 1 detailed the correlations between the synthetic and real clinical pathology data for each of the 38 measurements, demonstrating the small differences between AnimalGAN synthetic results and real laboratory animal testing data.

### AnimalGAN approach evaluation

The AnimalGAN approach was challenged with three training/test set split strategies to demonstrate its ability to produce reliable results for test drugs (1) whose chemical structures were far different from those that were used to build AnimalGAN itself, (2) whose therapeutical classes were not included in the development of AnimalGAN, and (3) that were approved by FDA more recently compared to the older drugs used to construct AnimalGAN. In all these three scenarios, the derived AnimalGAN models yielded the same results on the corresponding test sets as observed in the original AnimalGAN model. As summarized in Supplementary Fig. 2, the medians of cosine similarities between the synthetic and real data for all three scenarios were >0.99 (significantly higher than the median of background control 0.98 with $p$-values $< 5.65 \times 10^{-140}$) while the medians of RMSEs were <20.18 (significantly smaller than the median of background control 72.46 with $p$-values $< 4.80 \times 10^{-141}$). Additionally, in all these three scenarios, the

correlation between the synthetic and real data for each of the 38 clinical pathology measurements were comparable with those observed in the original AnimalGAN model (Supplementary Fig. 3). Here, the first extreme scenario was highlighted since there are concerns (e.g., applicability domain and activity cliff) when chemical information is the sole input in developing predictive models. In AnimalGAN, these concerns were mitigated by including the exposure information (i.e., dose and treatment duration) to warrant a robust application in real-world settings.

### AnimalGAN versus traditional AI approaches

We compared AnimalGAN results with QSAR analyses for each of 38 clinical pathology measurements. Specifically, for each measurement, we developed 12 regression models using the exact same study design and input (i.e., the descriptor and exposure information) as used in AnimalGAN; these are k-nearest neighbors, decision tree, extremely randomized tree, random forest, epsilon support vector regression, linear support vector regression, stochastic gradient descent, AdaBoost, gradient boosting, Bayesian ARD regression, Gaussian process regression and multi-layer perceptron (see Methods). As depicted in Fig. 2, AnimalGAN had much smaller Mean Square Error (MSE) between the predicted and true value than what can be achieved by all the QSAR models for every clinical pathology measurement (Supplementary Data 1). Of note, for each of 38 measurements, an individual QSAR model was developed while AnimalGAN generated the prediction for all 38 measurements at once.

### AnimalGAN application

A common scenario in toxicological assessment with animal data is to compare a measurement observed from a treatment group against its time-matched control group to determine a safety margin. We compared the AnimalGAN results with real animal testing data with this scenario (Fig. 3a and Methods), where a high agreement (i.e., 96.08% ~ 100%) was observed in the test set (see Supplementary Table 1). In both clinical and preclinical settings, out of the 38 clinical pathology measurements studied here, the seven are commonly used for hepatotoxicity assessment (i.e., alanine aminotransferase [ALT], aspartate aminotransferase [AST], lactate dehydrogenase [LDH], alkaline phosphatase [ALP], γ-glutamyltranspeptidase [GTP], total bilirubin [TBIL], and direct bilirubin [DBIL]) while the other seven are for nephrotoxicity assessment (i.e., blood urea nitrogen [BUN], creatinine [CRE], sodium [Na], potassium [K], chlorine [Cl], calcium [Ca], and inorganic phosphorus [IP]). The consistency between the AnimalGAN based assessment agreed with the animal studies for both hepatotoxicity and nephrotoxicity with the range of 96.08-100% and 97.89-100%, respectively (Fig. 3b, c), indicating the potential utility of AnimalGAN in animal-free testing (Fig. 3b, c).

### External validation with DrugMatrix data in toxicity assessment

An external validation of AnimalGAN was performed using a dataset derived from DrugMatrix[10]. It is a known that clinical pathology measurements can vary significantly between different experimental protocols or between different labs. For that reason, we analyzed the experiment data for the 70 common compounds (corresponding to 175 treatment conditions) tested in both TG-GATEs and DrugMatrix to establish a baseline concordance in their experiment settings. The overall average consistency between the two datasets for all 25 common measurements was 81.20%. For the external validation with 355 compounds under 717 treatment conditions, the consistency between AnimalGAN generated results and the real data from DrugMatrix was 82.85%. Figure 4a shows a comparison of the consistency of the baseline settings for all the 25 measurements based on experiment data and the AnimalGAN results. Moreover, we compared the chemistry space of 110 training compounds against the 355 external

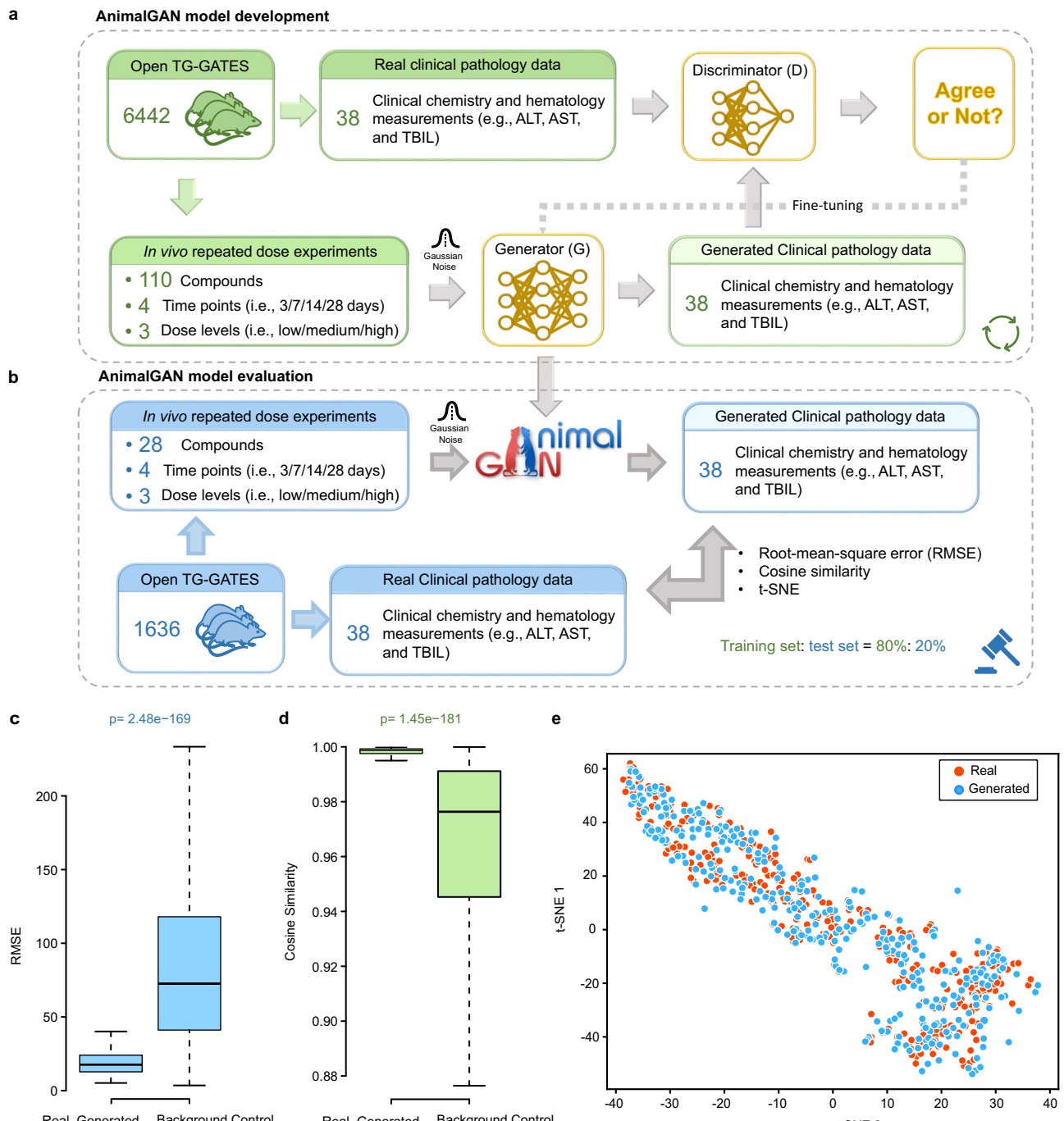

**Fig. 1 | AnimalGAN overview and study design. a** AnimalGAN model development. The AnimalGAN was developed based on 80% of TG-GATEs data (the training set) which consists of 6442 rats exposed to 110 compounds under 4 different time points (i.e., 3/7/14/28 days) and three dose levels (i.e., low/medium/high). The chemical representation (i.e., 1826 Mordred descriptors), time point, dose level, and Gaussian noise as input to the Generator (*G*) to yield the 38 synthetic clinical pathology measurements which was compared to the real data by the Discriminator (*D*). The average 100 generated clinical pathology measures passed the blood cell counts check to represent the clinical pathology measurements. Once the difference between the synthetic and real data could not be distinguished by the Discriminator (*D*), the AnimalGAN model was established. **b** AnimalGAN model evaluation. The AnimalGAN model was employed to generate the 38 clinical pathology measurements for 20% of TG-GATEs dataset (the test set) which consists of 332 treatment conditions exposed to 28 different compounds under 4 different time points (i.e., 3/7/14/28 days) and three dose levels (i.e., low/medium/high). We calculated the average 100 generated clinical pathology measures met a criterion

using the blood cell counts to represent the clinical pathology measurements from AnimalGAN and compared them to the corresponding real ones for each treatment condition. Boxplot of **c** RMSE - Root Mean Square Error and **d** Cosine Similarity between AnimalGAN generated synthetic data and real animal testing data for treatment conditions in the test set. The statistical difference between RMSEs/Cosine Similarities of AnimalGAN generated synthetic data and real animal testing data for *n* = 332 treatment conditions in the test set and RMSEs/Cosine Similarities of real data across any two treatment conditions (*n* = 1,358,776, derived from 1649 × 1648/2) was determined using a two-tailed Wilcoxon rank-sum test without adjustments for multiple comparisons. The boxplot displays the distribution of RMSEs/Cosine Similarities, with the centerline representing the median, the bounds of the box representing the first and third quantiles, and the whiskers representing the 1.5 times the interquartile range (IQR). **e** t-SNE visualization of generated data and real data for treatment conditions in the test set. Each point depicted one treatment condition. Source data are provided as a Source Data file.

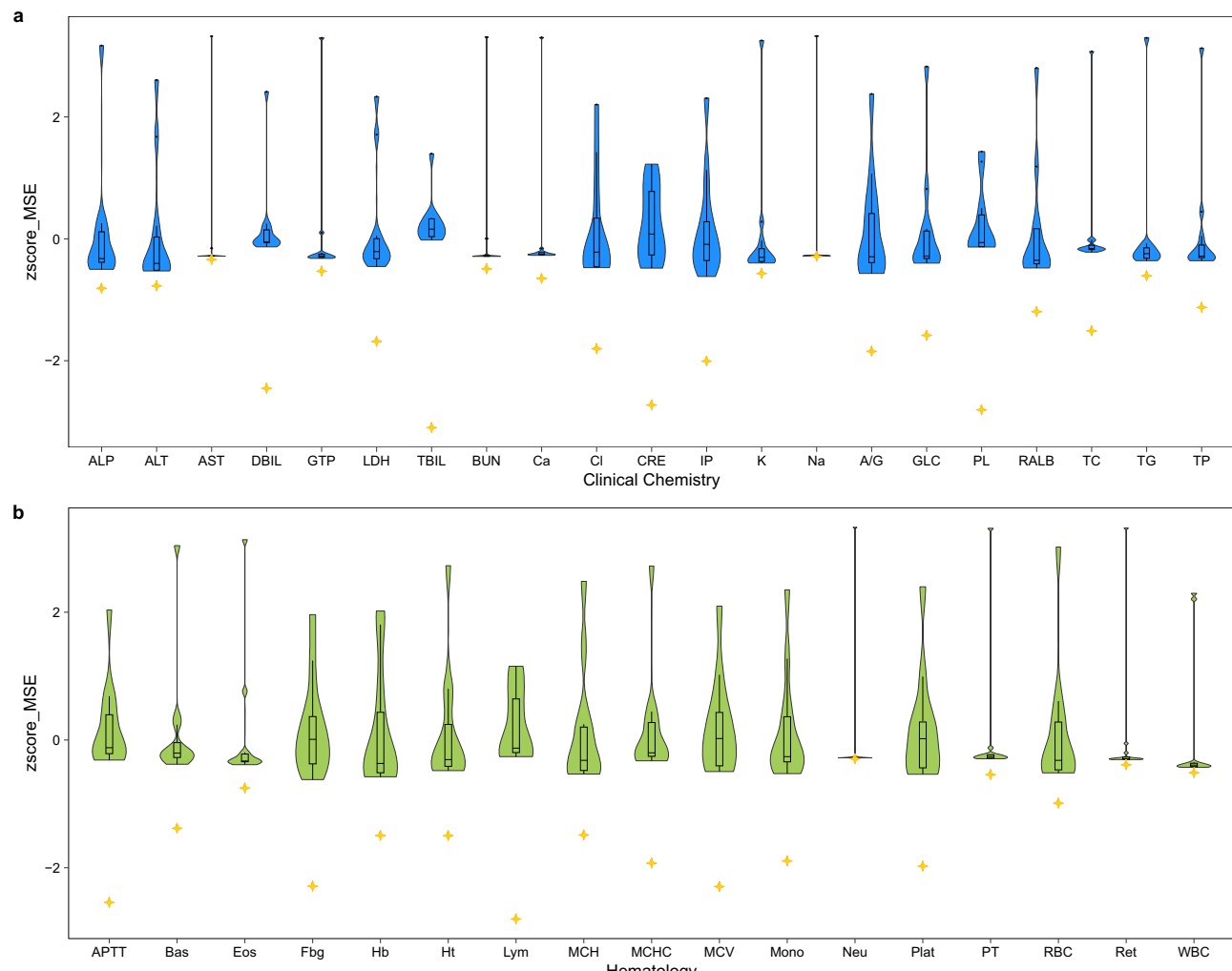

**Fig. 2 | Comparisons of AnimalGAN results with QSAR predictions for the test set of all 38 clinical pathology measurements. a** Clinical Chemistry measurements. ALP: alkaline phosphatase; ALT: alanine aminotransferase; AST aspartate aminotransferase, DBIL direct bilirubin, GTP γ-glutamyltranspeptidase, LDH lactate dehydrogenase, TBIL total bilirubin, BUN blood urea nitrogen, Ca calcium, Cl chlorine, CRE creatinine, IP inorganic phosphorus, K potassium, Na sodium, A/G albumin globulin ratio, GLC glucose, PL phospholipid, RALB albumin, TC total cholesterol, TG triglyceride, TP total protein. **b** Hematology measurements. APTT activated partial thromboplastin time, Bas basophil, Eos eosinophil, Fbg fibrinogen, Hb hemoglobin, Ht hematocrit value, Lym lymphocyte, MCH mean corpuscular hemoglobin, MCHC mean corpuscular hemoglobin concentration, MCV mean corpuscular volume, Mono monocyte, Neu neutrophil, Plat platelet count, PT

prothrombin time, RBC red blood cell count, Ret reticulocyte, WBC white blood cell count. For each measurement, the performance of the 12 QSARs was represented in a violin plot while the performance of AnimalGAN was denoted by a golden star. The plot was z-score scaled for an improved visual inspection. AnimalGAN exhibited consistently smaller MSE than what can be achieved with QSARs. The violin plot shows the distribution of the 12 QSARs' performance, with the width at different points indicating the estimated probability density of the data and the inner boxplot displaying the lower quantile (Q1), median (centerline), upper quantile (Q3) and whiskers extending to the minimum and maximum values within 1.5 times the interquartile range (IQR). All data points in this figure are provided in the Supplementary Data 1.

validation compounds and found that they were not significantly overlapped (Fig. 4b).

### AnimalGAN predicts idiosyncratic drug-induced liver injury (iDILI)

Since AnimalGAN is a virtual animal model, it could simulate the clinical pathology distribution from a large population of rats, from which the results might predict rare harmful events in the human population, thus offering an unprecedent opportunity to reliably translate the preclinical findings to clinical implications. For example, iDILI is rare and cannot be detected even in the late phase of clinical trials in drug development, let alone be foreseen in preclinical settings. Consequently, iDILI is only reported from the post-market surveillance and is thus a leading cause of drug recall and acute liver failure (ALF) in the United States[11]. Detecting iDILI has become one of the most challenging fields in pharmacovigilance, since the limited sample size in both

live animal and human studies are not suitable to provide sufficient statistical power. Here, we conducted a virtual 28-day study with AnimalGAN to generate liver enzyme data for a population of 100,000 rats under the treatment of high dose with each of three thiazolidinediones (i.e., troglitazone, pioglitazone, and rosiglitazone). The thiazolidinediones are a family of drugs with similar chemical structure for the treatment of type 2 diabetes[12]; troglitazone was withdrawn from the market due to its high frequency and severity of DILI, while pioglitazone and rosiglitazone are still on the market with less DILI frequency (less than 1%) and severity (most are mild and reversible). We examined the difference of these three thiazolidinediones in DILI risk by counting the number of rats (out of 100,000 simulation) that were above Upper Limit of Normal (ULN) in key liver enzymes that are traditionally used to assess DILI; these are ALT, AST and TBIL. ALT and AST measure the degree of liver injury while TBIL indicates the loss of liver function. We specifically emphasized on the classic Hy's law[13] that

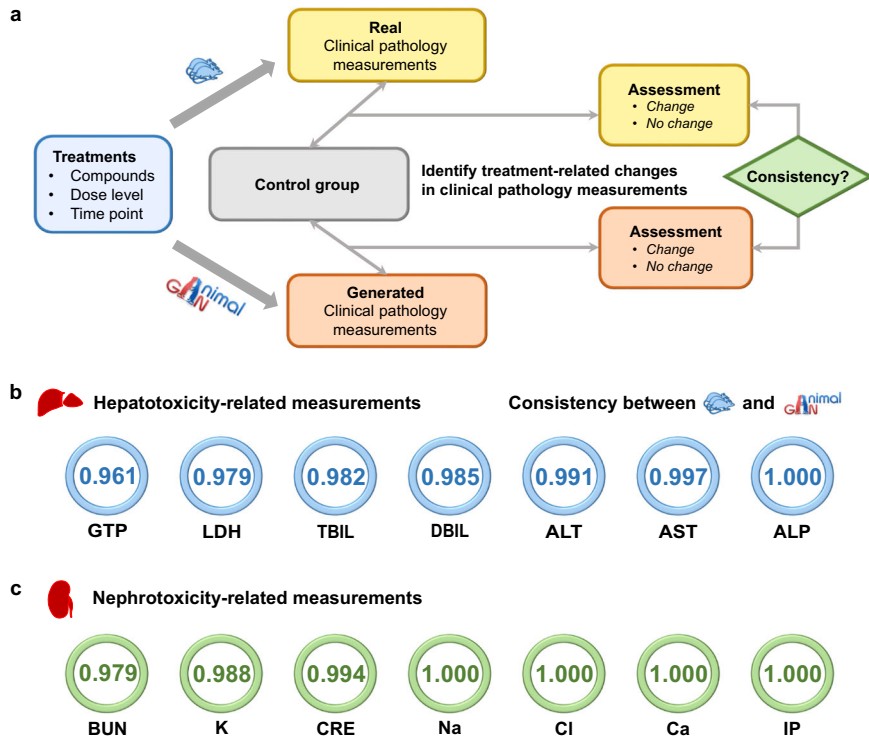

**Fig. 3 | AnimalGAN for toxicity assessment. a** A framework for comparing toxicity assessment outcomes between clinical pathology measurements generated by AnimalGAN and those from real rat experiments conducted under identical treatment conditions. Clinical pathology measurements were generated by AnimalGAN for each treatment condition (i.e., compound/time/dose). Then, each generated clinical pathology measurement and its corresponding real one (i.e., treated group) were analyzed against their matched controls to establish a statistically significant toxicity outcome. If both real and synthetic data lead to conclude the same toxicity outcomes, we consider that an agreement or "consistency" is established between experiment and AnimalGAN. The consistency for **b** hepatotoxicity and **c** nephrotoxicity-related clinical pathology measurements between generated data and their corresponding animal testing data in the test set. The consistency results for all the 38 clinical pathology measurements can be found in the Supplementary Table 1.

combines ALT (or AST) with TBIL and been traditionally used to assess the overall risk of DILI in clinical settings. As summarized in Table 1, Troglitazone had much more rats with liver enzyme elevation (except ALT) above their ULNs than these observed for both Pioglitazone and Rosiglitazone. Particularly, in terms of the overall DILI risk measured by the combination of ALT (or AST) with TBIL, the DILI frequency of Troglitazone was more than double than these from both Pioglitazone and Rosiglitazone.

## Discussion

Global efforts have been made to promote animal-free studies as alternatives to animal testing, such as the FDA Modernization Act[14], the FDA Predictive Toxicology RoadMap[15], the Tox21 program[16], and ONTOX[17] in Europe. Computational Toxicology has been actively evaluated as a part of this non-animal and 3Rs efforts. However, this field has remained unchanged for decades by predominantly relying on QSAR-like approaches. QSARs usually predict a single endpoint, often representing a high-level abstraction (e.g., liver toxicity vs non liver toxicity) of toxicological effects and as such are often missing critical contextual information. In contrast, animal-based assessments of toxicity provide much richer, multi-dimensional information to support risk assessment and decision making such as that derived from clinical pathology profiles. AnimalGAN represents a significant step towards synthetic, multi-dimensional toxicological profiles that reflect traditional preclinical toxicity assessments. By comparing to 12 conventional QSAR methods for each of 38 clinical pathology measures using the exact same input, AnimalGAN outperformed QSARs in prediction of all the clinical pathology measurements within the chemistry space pertaining to TG-GATEs. We acknowledge that AnimalGAN was developed within a relatively restricted chemical space,

which may impact the generalizability of our findings. The conclusion that AnimalGAN outperforms conventional QSARs is based on the dataset we used, and caution should be exercised when extending our results to diverse chemical classes or environmental compounds. It is important to point out that AnimalGAN is distinct from conventional QSAR approaches; the former can generate the prediction for the entire toxicological profiles (such as clinical pathology profile as reported here) at once while the latter predict one endpoint at a time. The significance of AnimalGAN not only lies in its predictive accuracy but also in its potential to replicate the complexity and richness of data obtained through conventional animal testing.

It is important to note that selecting a validation dataset for AnimalGAN is a challenge since different experimental study designs can lead to different test results. We established several criteria to select a reasonable validation set in our study: (1) same rat strain and sex, (2) a similar repeated dose study design as TG-GATEs, (3) common compounds tested by TG-GATEs to establish a baseline in comparison, and (4) contained clinical pathology measurements that significantly overlapped with those tested by TG-GATEs. DrugMatrix met all these criteria with several minor deviations. The first consideration is about the age of rats; DrugMatrix used rats of 5–7 weeks while TG-GATEs used rats that were 6-weeks, which might not be significant but worth to mention. The second consideration is about the testing strategy to determine Maximum Tolerated Dose (MTD); DrugMatrix applied a 5-day dose range finding method while TG-GATEs worked on a 7-day protocol, which might also not result in significant difference between two studies. The third consideration is in defining the dose level, which might have a significant impact; DrugMatrix applied both MTD and therapeutic dose to define dose levels whereas TG-GATEs only relied on MTD. The fourth consideration regards control groups; DrugMatrix

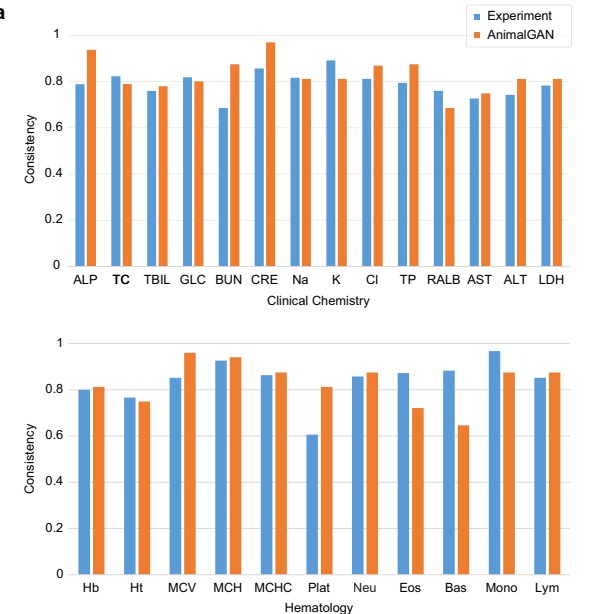

**Fig. 4 | External validation of AnimalGAN. a** Comparison of Consistency in Toxicity Assessment Outcomes based on experiment data and AnimalGAN-generated data for all 25 measurements. The blue (left) bars represent the consistency in toxicity assessment outcomes between the experiment data from TG-GATEs and DrugMatrix. The orange (right) bars represent the consistency in toxicity assessment outcomes between the AnimalGAN-generated measurements and the experiment data tested in DrugMatrix. **b** Visualization of chemistry space of the 110 training compounds and the 355 external validation compounds. Each point depicts one compound. Source data are provided as a Source Data file.

used a shared control while TG-GATEs applied time-matched controls. In addition, the source and purity of compounds might be different between two datasets, which we do not have information. Considering these differences between the two study designs, around 81% consistency in toxicity assessment for the common compounds using the experimental data seems reasonable. Consequently, the observed ~83% consistency of AnimalGAN results against the DrugMatrix experiment data demonstrates its potential application for toxicity assessment.

It is widely acknowledged that Phase III trials in drug development may not reliably predict rare adverse events, such as iDILI. This is mainly due to the limited and controlled population sample in these trials, which may not be representative of the real-world population. However, when it comes to explaining the poor translation of animal study results to humans, the main factor often cited is species differences in biology. Despite this, we speculate that the small sample size used in animal studies may also contribute to the poor translation. Since AnimalGAN is a virtual experiment, we carried out a 28-day study with a large population of rats for DILI assessment of three thiazolidinediones. These three drugs are similar in chemical structure (share the same scaffold) but with different DILI risk. The virtual experiment with 100,000 rats revealed the difference in DILI potential

among these three. If we estimated the DILI frequency as a percentage of rats meeting the overall DILI risk criteria (the last row of Table 1), the AnimalGAN results agreed with the iDILI frequencies in the human population of these three thiazolidinediones; that is 1.9% (troglitazone), 0.26% (pioglitazone), 0.25% (rosiglitazone). The results offer a potential venue to assess rare adverse events in the human population which are unlikely to be detected in conventional animal studies.

While several less complex toxicological endpoints have found adequate non-animal replacements, many efforts in this space have run up against the challenge presented by the diversity and complexity of systemic toxicological processes necessary to characterize toxicity and establish safety. In addition, there are around 4,000 new chemicals per day in U.S., making testing via conventional animal studies an impossible task. Even some proposed alternative methods, such as micro-physiological systems (e.g., organ-on-a-chip), do not possess the throughput capacity to deal with such a large number of compounds to be tested[18]. Therefore, read across has been introduced to address this challenge by using existing chemical toxicity information to predict untested compounds based on chemical structure similarity. However, the assumption of chemical-based read-across is not always correct since chemicals with similar structures could have different bioactivities and carry different hazards. AnimalGAN could overcome these challenges by providing synthetic results for a large number of chemicals, thus improving read-across.

It is worthwhile to point out that AnimalGAN was only evaluated in a fraction of toxicological space in its current form. Hence, a broad assessment with much larger number of rats could further improve its robustness and lead to a model to supplement toxicological assessment in animals. Nonetheless, in the short term, AnimalGAN provides the opportunity to generate data that will allow for rational prioritization of drugs based on toxicological risk. Looking further into the future, it is anticipated that AI learning from more comprehensive sets of animal data will allow for more comprehensive forecasting of toxicological effects and chemical and drug safety assessment without the use of animals.

**Table 1 | The number of rats exhibiting drug-induced liver injury estimated by AnimalGAN for the three thiazolidinediones under the 28-day study with high dose in 100,000 rats**

| Criteria | Troglitazone | Pioglitazone | Rosiglitazone |
|---|---|---|---|
| ALT > ULN[a] | 1230 | 1820 | 1467 |
| AST > ULN | 7413 | 4315 | 4591 |
| TBIL > ULN | 3421 | 2083 | 2215 |
| ALT > ULN or AST > ULN, and TBIL > ULN | 375 | 161 | 158 |

[a]The Upper Limit of Normal (ULN) was defined as the highest measurement value observed in the 28-day vehicle-treated control rats.

## Methods

### Clinical pathology data from open TG-GATEs

The Open TG-GATEs[9] was employed to develop the AnimalGAN model. The Open TG-GATEs is a large-scale publicly available toxicological research database that stores toxicogenomic profiles and traditional toxicological data derived from in vivo (i.e., rat liver and kidney) and in vitro (i.e., rat and human primary hepatocyte) exposure to 170 compounds at multiple dosages and time points. In this study, we focused on 38 clinical pathology measurements, including 21 hematology and 17 biochemistry datapoints, generated using the standard animal study protocols with three dose levels (low, middle, and high) and four treatment durations (3, 7, 14, and 28 days). All the data were downloaded from the website https://dbarchive.biosciencedbc.jp/en/open-tggates/download.html, accessed in September 2021. Clinical pathology data of 8078 rats treated with 138 compounds under 1649 treatment conditions (i.e., compound/duration/dose combinations) were used for model development and validation. In addition, clinical pathology data from 2775 vehicle control-treated rats were downloaded as controls for toxicity assessment. The detailed information on treatment conditions is listed in Supplementary Data 2.

### Molecular representation

Chemical structure was one of the inputs used to develop AnimalGAN, which was represented with numeric molecular descriptors. First, compound information for the 138 compounds, including PubChem CID, Structure-Data File (SDF) and canonical simplified molecular-input line-entry system (SMILES), was retrieved from the PubChem database[19] using PubChemPy[20] (Supplementary Data 3). Then, molecular representations were calculated using a Python package, Mordred[21], generating a length of 1826 2D and 3D molecular descriptors for each compound.

### Overview of AnimalGAN model

The architecture of the AnimalGAN is illustrated in Fig. 1a, Supplementary Fig. 4 and Supplementary Table 2. The AnimalGAN model comprises a generator $G$ and a discriminator $D$. The generator $G$ is fed with a conditional input $c$ (a combination of compound, time, and dose) and a random input $z$ to generate the simulated clinical pathology measurements $\tilde{x}$. The discriminator $D$ takes real and generated clinical pathology measurements under the condition $c$ as the input and analyzed their difference. During the training process, the generator and the discriminator compete against each other and thus improve each other iteratively. At each training step, as the generator $G$ tried to produce hematologic and clinical chemistry data similar to the real data, the discriminator $D$ became better at distinguishing between real data from animals and generated data produced by the generator. The model was considered converged when the discriminator could not distinguish the synthetic results from the real ones. By then, the AnimalGAN model was finalized and could be used to infer clinical pathology measurements of rats treated with untested compounds at a specific time and dose.

GANs are a powerful class of generative models that aim to learn a mapping from input distribution to output distribution, in this case, a mapping from animal study conditions (the combination of chemical, dose and treatment duration) to clinical pathology measurements. The vanilla GAN[8] structure comprises two neural networks: a generator $G$ and a discriminator $D$ iteratively trained by competing against each other in a min-max game with the following learning objective:

$$\min_G \max_D \mathcal{L}(D, G) = \min_G \max_D \{\mathbb{E}_{x \sim \mathcal{P}_{real}}[\log D(x)] + \mathbb{E}_{z \sim \mathcal{P}_z}[\log(1 - D(G(z)))]\} \quad (1)$$

where $\mathbb{E}[\cdot]$ represents expectation, $x$ is a vector of the clinical pathology measurements sampled from the distribution of real lab test reports $\mathcal{P}_{real}$, and $z$ is a vector with random noise sampled from a Gaussian distribution $P_z$.

Conditional GAN (cGAN)[22] is developed to generate data with the specific condition. In the cGAN, both generator and discriminator have the condition as one of the inputs, and the loss function is expressed as

$$\mathcal{L}(D, G) = \mathbb{E}_{x \sim \mathcal{P}_{real}}[\log D(x|c)] + \mathbb{E}_{z \sim \mathcal{P}_z}[\log(1 - D(G(z|c)))] \quad (2)$$

While cGAN has demonstrated outstanding capabilities in a wide range of conditional generation tasks, training cGAN as playing a min-max game between the generator network and discriminator network is inherently unstable, especially for biomedical data with small size training samples, and the conditions are continuous and infinite and high-dimensional. Wasserstein-GAN (WGAN)[23] employed the Wasserstein distance, also called the Earth Mover (EM) distance, as a more explicit measure of the distribution divergence in the loss function to overcome the gradient disappearance to partially alleviate the mode collapse. Herein, the AnimalGAN framework was developed based on the combination of cGAN and WGAN. cGAN allowed the generation of clinical pathology measures at a given condition (i.e., compound/time/dose), while WGAN improved the convergence of the model. Furthermore, due to the small sample size and continuous high-dimensional conditions in clinical pathology data, a regularization term was implanted on generator loss to improve the generalizability of the generator, allowing the generator to leverage neighboring conditions in the continuous space without sacrificing the generator's faithfulness to the input conditions through Lipschitz regularization along with the interpolations of conditions pairs. When the generator was fed with new conditions that have never been seen in the training set, the regularization term then leverages the neighbor information and assists the generator in generating data that have similar conditional distributions for each neighboring condition. Specifically, the following regularization term was added to the generator loss to encourage the optimized generator $G$ to minimize this regularization term:

$$\mathcal{L}_{GR}(G) = \mathbb{E}_{z \sim \mathcal{P}_z, c \sim \mathcal{P}_c}\left[\min\left(\frac{|G(c + \triangle c, z) - G(c, z)|}{|\triangle x|}, \tau\right)\right] \quad (3)$$

where $\triangle c \sim \mathcal{P}_{\triangle c}$ is a small perturbation added to $c$ and $\mathcal{P}_{\triangle c}$ is the distribution of $\triangle c$. The distribution $\mathcal{P}_{\triangle c}$ was designed to be a distribution centered close to zero with small variance and normal distribution. $\tau$ is a bound for ensuring numerical stability.

Finally, the model objective with generator regularization becomes

$$\min_G \max_D \mathcal{L}(D, G) = \mathbb{E}_{(c,x) \sim \mathcal{P}_{(c,x)}}[\log D(c, x)] \\ + \mathbb{E}_{z \sim \mathcal{P}_z, c \sim \mathcal{P}_c}[\log(1 - D(G(c, z)))] + \lambda \mathcal{L}_{GR}(G) \quad (4)$$

The condition space $c$ in this study is the treatment information (i.e., compound/duration/dosage combinations), and the output space is the clinical pathology measurements.

### Generator architecture

The generator $G$ received four inputs, with the first three inputs as the treatment condition $c = concat(s, d, t)$. The first one was the molecular representation of a chemical ($s$), an 1826-dimensional vector of molecular representation by using Mordred. The second input was the dose level of a compound administered to the rat ($d$). For the animal studies in the Open TG-GATEs database, the ratio among low, middle, and high dose levels was set as 1:3:10 with the high dose equal to the Maximum Tolerance Dose determined in a 7-day dose finding study, with the same setting applied here. The third input denotes the time point (3, 7, 14, and 28 days) a rat ($t$) was treated. The final input was an 1828-dimensional noise vector ($n$) sampled from a normal distribution. All these four inputs were concatenated into a 3656-dimensional

vector and scaled to [−1, 1] as input to pass into the generator $G$, resulting in $\tilde{x} = G(c, z)$, where $\tilde{x}$ was a generated vector of clinical pathology measurements. The generator $G(c, z)$ was a fully connected neural network with 5 hidden layers, where each layer has 4096, 2048, 1024, 256 and 64 nodes, respectively, and used LeakyReLU as activation function, followed by a batch normalization procedure. The output layer has 38 nodes, which is equal to the dimensionality of the clinical pathology measurements in the Open TG-GATEs database, followed by a tanh activation function.

### Discriminator architecture

The 38-dimensional vector of clinical pathology measurements and 1828-dimensional vector of treatment conditions (i.e., descriptors + dose + time) were concatenated into an 1866-dimentional vector and scaled to [−1,1] as an input to the discriminator. The discriminator was a seven-layer multilayer perceptron (MLP) of hidden layers with 2048, 1024, 256, 64, and 32 neurons, respectively. We used the activation function LeakyReLU in all the hidden layers with the angle of the negative slope of 0.2. To avoid overfitting, we used dropout with a rate of 0.4 after each hidden layer.

### AnimalGAN model development

Of clinical pathology data consisting of 8078 rats treated with 138 compounds under 1649 treatment conditions, 38 clinical pathology measurements (i.e., 21 hematologic and 17 clinical chemistry) of 110 compounds (~80%), corresponding to 6442 rats under 1317 treatment conditions were randomly selected as a training set. The 21 hematologic and 17 clinical chemistry measurements of the 1636 rats under 332 treatment conditions of the remaining 28 compounds (~20%) were used as a test set.

 Using the proposed loss function above, the converged Animal-GAN model could only generate the clinical pathology measurements with a similar distribution to the real ones. To enable the Generator $G$ to generate the clinical pathology measurements maximally identical to the real ones, additional post optimization criteria were proposed. First, an "invalid records" check based on the blood cell counts was implemented. Among the 38 clinical pathology measurements, white blood cells (WBCs) are composed of neutrophils, eosinophils, basophils, monocytes, and lymphocytes, so the total percentages of each type of WBC should not exceed 100%. Considering the cell counting protocol performed in the lab and the rounding numbers used in test results, the cell count with total percentages of more than 105% were considered invalid records. In other words, only the generated clinical pathology measurements that passed the blood cell counts check were kept. Then, cosine similarity and RMSE were ensured between generated clinical pathology measurements and their corresponding real ones to be optimum, where the cosine similarity and RMSE of a given treatment condition are defined as below:

$$cosine\,similariy = \frac{\sum_{i=1}^{38} \bar{G}_i \bar{R}_i}{\sqrt{\sum_{i=1}^{38} \bar{G}_i^2} \sqrt{\sum_{i=1}^{38} \bar{R}_i^2}} \quad (5)$$

$$RMSE = \sqrt{\frac{\sum_{i=1}^{38} (\bar{G}_i - \bar{R}_i)^2}{38}} \quad (6)$$

where $\bar{G}_i$ and $\bar{R}_i$ are the $i$-th measurement of the mean vector of generated clinical pathology results and their corresponding real ones, respectively.

 The generator $G$ and discriminator $D$ losses tended to be stable after 1000 epochs, respectively. To further optimize the AnimalGAN, the AnimalGAN models were examined in different checkpoints using the proposed post-optimization strategy mentioned above. Using the stored AnimalGAN model under each training epoch, the average 100 generated clinical pathology measures that passed the blood cell counts check were calculated to represent the clinical pathology measurements in the training set and compared to the corresponding real ones for each treatment condition (Supplementary Fig. 5). The cosine similarity and RMSE between generated clinical pathology measures and their corresponding real ones in the training set reached the optimal at 6000 epochs. Also, the similar distributions of generated clinical pathology measurements and their corresponding real animal data in the training set in $t$-distributed stochastic neighbor embedding ($t$-SNE) space suggested that the AnimalGAN model at 6000 epochs was well-trained.

### AnimalGAN predicts clinical pathology measurements

The test set consisting of the 1636 rats under 332 treatment conditions of the 28 different compounds were used to evaluate the AnimalGAN performance, where the treatment conditions (i.e., compound/time/ dose) were input to AnimalGAN to generate clinical pathology data. For each treatment condition, the averaged 100 generated clinical pathology measures that passed the blood cell count check were employed to represent the AnimalGAN synthetic testing results, followed with additional verifications using the cosine similarity and RMSE between the averaged clinical pathology data from AnimalGAN and their corresponding real ones under the same treatment condition, respectively. Furthermore, the proposed control analysis[24] in our previous study was utilized to justify whether the similarities between the generated clinical pathology measurements and their corresponding real ones were superior to those of background control distribution, where the similarities were generated between real clinical pathology measurements in any two treatment conditions. Specifically, in order to construct a background control distribution, we calculated the Cosine Similarities and RMSEs between every pair of treatment conditions (i.e., compound/time/dose combination) based on their lab-tested clinical pathology profiles. We then compared the distributions of Cosine Similarities and RMSEs between the profiles generated by AnimalGAN and the corresponding profiles obtained from TG-GATEs against the background control distributions. We employed Wilcoxon rank-sum tests to compare these distributions against the background control distributions. The $t$-SNE was also used to visualize the generated data and real data.

### AnimalGAN evaluation

To investigate the applicability domain of the proposed AnimalGAN, we employed three scenarios to mimic real-world situations by repartitioning the training set and test set. To warrant a fair comparison among the different scenarios, we kept the same ratio (80% and 20%) of compounds in the training set (i.e., 110 compounds) and test set (i.e., the remaining 28 compounds). The three designed scenarios are listed below:

(1) To investigate whether the AnimalGAN model could infer the clinical pathology measures for the compounds that are not similar to those included in the training process, the pairwise structural similarities between any two of the 138 compounds were first calculated based on their Mordred molecular representations. Second, the compounds were ranked according to their median similarities to others, and then the sorted compounds were partitioned into two sets, with the first 110 compounds in one set, and the last 28 compounds in anther set. Via this, treatment conditions using the 110 compounds with higher similarities to others were used as training set to develop model and treatment conditions using the remaining 28 compounds with little structural similarity to the compounds in the training set were used as test set to evaluate the model (Supplementary Figure 6 and Supplementary Data 2).

(2) Whether the AnimalGAN framework could be used to infer clinical pathology measurements for drugs whose therapeutic use

was not included in the training set was also investigated. Specifically, the 138 compounds were mapped onto the first level of the WHO Anatomical Therapeutic Chemical (ATC, https://www.whocc.no/atc_ddd_index/) code, which represents which organ that drug impacts. Then, the AnimalGAN model was developed with drugs from therapeutic categories, including A- Alimentary tract and metabolism, H- Systemic hormonal preparations, excluding sex hormones and insulins, L- antineoplastic and immunomodulating agents, along with compounds with ATC code. Then, drugs belonging to R- respiratory system, and P- antiparasitic products, insecticides, and repellents were left as the test set to evaluate the developed AnimalGAN (Supplementary Data 2).

(3) A 'time-split' strategy[7] was employed to investigate whether AnimalGAN could be used to infer clinical pathology measurements based on accumulated animal data to foresee the animal response of untested compounds. For that, drugs approved before the year 1982 and non-drug-like compounds were used as the training set to develop AnimalGAN, while drugs approved from the year 1982 onward were held out as the test set to examine the performance of the model (Supplementary Data 2).

Similarly, the cosine similarity and RMSE between the generated clinical pathology measurements and their corresponding real ones were used to evaluate the model performance. Also, prediction errors of synthetic results against actual laboratory animal testing values were also analyzed for each of the 38 clinical pathology measurements.

## Comparing AnimalGAN results with QSAR predictions

QSARs are computational toxicology methods that are widely used in chemical and pharmacological research to predict the biological and toxicological activity of a compound based on its structural features. To compare the performance of AnimalGAN with QSARs, we used the same input for both approaches. Specifically, for each of the 38 clinical pathology measurements, 12 regressors (i.e., k-nearest neighbors, decision tree, extremely randomized tree, random forest, epsilon support vector regression, linear support vector regression, stochastic gradient descent, AdaBoost, gradient boosting, Bayesian ARD regression, Gaussian process regression and multi-layer perceptron) were used to develop QSAR models. To validate the QSARs models, we used the same test set as used in AnimalGAN for a fair comparison. In addition, the same molecular descriptors along with the exposure information (i.e., dose and treatment duration) used in AnimalGAN were the input to QSARs modeling. A 5-fold cross-validation was used for hyperparameter optimization, followed with the final model construction using the entire training set. Then, the predictions for the test set were made by the QSAR models and then subsequently compared with AnimalGAN results using Mean Square Error (MSE) which measures the difference between the predicted value and true value.

## AnimalGAN for toxicity assessment

Clinical pathology measurements are critical to assess the toxicity of different organ systems in the preclinical setting. To demonstrate the potential application of AnimalGAN in toxicity assessment, we carried out a comparison of toxicity assessment outcomes derived from clinical pathology measurements generated by AnimalGAN and real rat experiments conducted under the same treatment conditions. Clinical pathology measurements were generated by AnimalGAN for each treatment condition (i.e., compound/time/dose). Then, each generated clinical pathology measurement and the corresponding real one (i.e., treated group) were analyzed against their matched controls to establish a statistically significant toxicity outcome. Specifically, unpaired $t$-tests were performed to compare the control and treated groups, and values of $p < 0.05$ were considered statistically significant. Calculations were based on the hypothesis that the clinical pathology

measurements are normally distributed and that the within-group variances are the same. For those measurements that did not follow the normal distribution, non-parametric tests were used. Eventually, if both real and synthetic data lead to conclude the same toxicity outcomes, we consider that an agreement or "consistency" was established between experiment and AnimalGAN, otherwise, an inconsistent toxicity outcome was concluded. The consistency was calculated based on the following formula,

$$consistency_i = \frac{Number\ of\ treatment\ conditions\ with\ consistent\ toxicity\ outcomes}{Total\ of\ treatment\ condidtions}$$

(7)

where $i$ denotes the $i$-th clinical pathology measurement.

## External validation with DrugMatrix dataset

The other largest, publicly available, TG-GATEs comparable dataset is DrugMatrix[10]. This is managed by the National Toxicology Program of the US Department of Health and Human Services and was used for external validation in this study. The dataset includes 38 clinical pathology measurements, 25 of which overlap with those in TG-GATEs. Subsequently, these 25 common clinical pathology measurements were evaluated to assess AnimalGAN performance on the DrugMatrix data. The animal testing data with treatment conditions from DrugMatrix was downloaded from ftp://anonftp.niehs.nih.gov/ntp-cebs/datatype/Drug_Matrix/DrugMatrixPostgreSqlDatabase.tar.gz. To ensure that the study design for animal data from DrugMatrix was comparable to TG-GATEs, the following animal data from DrugMatrix were excluded: (1) compounds that did not have a matched vehicle control, (2) samples where a different route of administration was used between treated and control conditions, (3) samples where the treated and control experiments were conducted in different labs, or (4) samples where studies were from female rats. As a result, 355 compounds with 717 treatment conditions (i.e., compound/dose/time combination) were used as an external dataset, which were not tested in TG-GATEs (Supplementary Data 4). In addition, there were 70 compounds, corresponding to 175 treatment conditions, which were tested in both TG-GATEs and DrugMatrix. We analyzed the concordance between the two databases for these 70 compounds to serve as a baseline to assess the external validation results on 355 compounds.

The external validation was carried out as described in the previous section (i.e., AnimalGAN Application). Specifically, for each of the 717 treatment conditions of the external validation set, 100 simulations with valid blood records were generated with AnimalGAN and were subsequently compared against matched controls to calculate the toxicity of each of the 25 clinical pathology measures. The toxicity outcome for each measurement was then compared to the outcome from the real experimental results in DrugMatrix to calculate consistency as an external validation of the performance of AnimalGAN.

## Reporting summary

Further information on research design is available in the Nature Portfolio Reporting Summary linked to this article.

# Data availability

The datasets used to develop and evaluate AnimalGAN are sourced from clinical pathology data in the Open TG-GATEs database, available for downloaded at https://dbarchive.biosciencedbc.jp/en/open-tggates/download.html. The PubChem database can be accessed at https://pubchem.ncbi.nlm.nih.gov/. The animal testing data with treatment conditions from DrugMatrix were downloaded from ftp://anonftp.niehs.nih.gov/ntp-cebs/datatype/Drug_Matrix/DrugMatrixPostgreSqlDatabase.tar.gz. Source data are provided with this paper.

## Code availability

All the networks were built and trained using PyTorch version 1.11.0 with CUDA 10.2 under open-source Python (version 3.9.18). The source code is available at https://github.com/XC-NCTR/AnimalGAN, which has also been deposited in the Zenodo at https://doi.org/10.5281/zenodo.8416406[25].

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

## Acknowledgements

We thank Dr. Scott Auerbach from Division of the Translational Toxicology, National Institute of Environmental Health Sciences (Research Triangle Park, North Carolina, USA) for inspiring discussions during the preparation of this manuscript. This manuscript reflects the views of the authors and does not necessarily reflect those of the Food and Drug Administration. Any mention of commercial products is for clarification only and is not intended as approval, endorsement, or recommendation.

## Author contributions

W.T. and Z.L. conceived and designed the study. X.C. collected the datasets, performed the model development and computational analysis. Z.L., X.C. and W.T. wrote the manuscript. X.C., W.T. and R.R. revised the manuscript. All authors edited and approved the final manuscript. Z.L. conducted this work when he worked at the National Center for Toxicological Research, Food and Drug Administration.

## Competing interests

R.R. is co-founder and co-director of ApconiX, an integrated toxicology and ion channel company that provides expert advice on non-clinical aspects of drug discovery and drug development to academia, industry, and not-for-profit organizations. The remaining authors declare no competing interests.
