## [Peer Review File · Nature Communications]

A Generative Adversarial Network Model Alternative to Animal Studies for Clinical Pathology AssessmentREVIEWER COMMENTS

Reviewer #1 (Remarks to the Author):

The aim of this study is to develop a GAN method to simulate 38 rat clinical pathology measures and evaluate from significantly different drugs to training in terms of chemical structure, drug class, and the year of FDA approval. This is quite interesting and one of important topics. However there are several concerns on the reproducibility.

1. This test data was split by open TG-GATEs, which lead to a lack of external validation.
2. Why are 1649 tx conditions? Is it $138 * 4 * 3$ (comp/duration/dose) = 1656? Please describe it in detail.
3. Please summarize relevant GAN in M&M with references
4. Is there any missing in duration? How do you deal with imbalances of dose?
5. Please present a figure of architecture of generator and discriminator.
6. Cross validate component splitting with k-folding and present Standard deviation.
7. What's all these four inputs with 3656?
8. There is a lack of parameter and model searching, and ablation studies.
9. There should be Bland Altman analysis on 38 measurements, which need to be evaluated in viewpoint of systemic bias.
10. Fig 2 should be tabulated with statistical comparison.

Reviewer #2 (Remarks to the Author):

In their article titled "AnimalGAN: A GAN Model for Clinical Pathology Assessment," Chen et al. propose the use of Generative Adversarial Networks (GANs) as an alternative to animal studies in evaluating chemical and drug safety. Their model, AnimalGAN, successfully simulated 38 rat clinical pathology measures and showed comparable results to real animal data in hepatotoxicity assessment, surpassing conventional approaches. AnimalGAN's virtual experiment accurately ranked the hepatotoxicity of three structurally similar drugs, demonstrating its potential for reducing and refining animal use in research.

The paper is well-written and easily understandable, with high-quality display items that effectively enhance the clarity of the information presented.

How does the evaluation of the model's performance on the test set demonstrate its effectiveness and practicality for real studies? In addition to reporting the mean squared error (MSE), I am interested in understanding how the model performs under an ablation study. Furthermore, I would suggest interpolating the molecular descriptor between DSMO and known drug-induced liver injury (DILI) compounds to observe how the clinical pathology findings change. It would be valuable to keep the noise pattern schedule fixed for the intermediate state during this analysis.

The authors should provide a more detailed description of how the study conditions were encoded and how the clinical pathology findings were normalized across different conditions to correct for plain batch effects. Regarding the evaluation, I would like to see the mean squared error (MSE) plotted over i) time points (3,7,14 and 28) after treatment and ii) dose level to ensure that the model is not learning trivialities that primarily minimize the overall MSE by focusing on the time or dose component.

The choice of input representation appears somewhat arbitrary. Have other molecular descriptors been tested as well? Additionally, what impact does the size of the 1828-dimensional noise vector have on the model's performance?

I see great potential in this work, and the chosen topic is highly relevant. However, I believe that there is room for improvement in the manuscript as it stands. Therefore, I request a revision to address the identified areas for improvement.

Reviewer #3 (Remarks to the Author):

The article presents the development of a new approach methodology (NAMs) using AI to simulate rats and predict 38 clinical pathology measurements, including 21 hematologic and 17 clinical chemistry parameters. As a training set, the authors used a subset of in vivo experiments available on the Open TG-GATEs platform. The dataset used for training included 8,078 rats treated with 138 compounds under 1,649 treatments. The model was validated by comparing it with QSAR models developed for the same purpose. Finally, a case study was conducted for three thiazolidinediones, and a population of 100,000 rats was generated.

The approach developed here is novel and leverages modern AI techniques to address toxicology problems. The paper is well-written, and the authors have made an effort to describe the methods in the journal format. Unfortunately, my review is limited to the content of the publication, as the links to the code and data (<https://github.com/XC-NCTR/AnimalGAN> , <https://dbarchive.biosciencedbc.jp/en/opentggates/download.html>) are not working or are empty. The authors should provide the following information for a more detailed review:

The methods for developing the AI model involved grid optimization, cosine similarity, and RMSE scoring. However, the authors did not discuss the issue of overfitting in the model. I was surprised to see a performance gap between epoch 5000 and epoch 6000 during the training phase, as well as almost perfect prediction error plots. The authors should address the issue of overfitting and attempt to estimate its impact.

Another concern I have is regarding the comparison with QSAR models. The supporting information document shows a list of chemicals with similarity scores, and almost all of them have scores above 0.9. If this is the case for the 138 compounds used in the training, it suggests that QSAR models built from molecular descriptors may not perform well due to the limited diversity of the chemicals. The authors should take this into consideration and discuss the limitations of their validation.

Furthermore, the authors should discuss the applicability of this model. How far can the model be extended with only 138 chemicals considered as input, and what considerations should future users have when utilizing this model? The issue of chemical diversity should also be addressed. If I understand the table in the supporting information correctly, it indicates poor diversity.

Minor comments:

- Could the authors provide more details on how they calculated the background control?
- Please check the author list, as there are differences between the manuscript and the supporting information.
- In Figure S6, please add the percentage of variance explained for each principal component in the PCA. Also, comment on the poor overlap between the training and test sets. Were the test samples randomly selected or optimized?
- I found it difficult to read and compare the performances in Figure 2. Using a non-circular figure would make it easier to read. Additionally, the square representing the performance of AnimalGAN could be improved.

REVIEWER COMMENTS

Reviewer #1 (Remarks to the Author):

The aim of this study is to develop a GAN method to simulate 38 rat clinical pathology measures and evaluate from significantly different drugs to training in terms of chemical structure, drug class, and the year of FDA approval. This is quite interesting and one of important topics. However there are several concerns on the reproducibility.

1. This test data was split by open TG-GATEs, which lead to a lack of external validation.

We thank the reviewer for this comment. In the revised version, we have included three new sections to address this point, one under M&M, one under Results and the other under Discussion (see below). We have also added several sentences to the Discussion to ensure this point is clear.

Materials and Methods:

External validation with DrugMatrix dataset

The other largest, publicly available, TG-GATEs comparable dataset is DrugMatrix[9]. This is managed by the National Toxicology Program of the US Department of Health and Human Services and was used for external validation in this study. The dataset includes 38 clinical pathology measurements, 25 of which overlap with those in TG-GATEs. Subsequently, these 25 common clinical pathology measurements were evaluated to assess AnimalGAN performance on the DrugMatrix data. The animal testing data with treatment conditions from DrugMatrix was downloaded from ftp://anonftp.niehs.nih.gov/ntp-cebs/datatype/Drug_Matrix/DrugMatrixPostgreSQLDatabase.tar.gz. To ensure that the study design for animal data from DrugMatrix was comparable to TG-GATEs, the following animal data from DrugMatrix were excluded: (1) compounds that did not have a matched vehicle control, (2) samples where a different route of administration was used between treated and control conditions, (3) samples where the treated and control experiments were conducted in different labs, or (4) samples where studies were from female rats. As a result, 355 compounds with 717 treatment conditions (i.e., compound/dose/time combination) were used as an external dataset, which were not tested in TG-GATEs (**Supplementary Table S6**). In addition, there were 70 compounds, corresponding to 175 treatment conditions, which were tested in both TG-GATEs and DrugMatrix. We analyzed the concordance between the two databases for these 70 compounds to serve as a baseline to assess the external validation results on 355 compounds.

The external validation was carried out as described in the previous section (i.e., AnimalGAN Application). Specifically, for each of the 717 treatment conditions of the external validation set, 100 simulations with valid blood records were generated with AnimalGAN and were subsequently compared against matched controls to calculate the toxicity of each of the 25 clinical pathology measures. The toxicity outcome for each measurement was then compared to the outcome from the real experimental results in DrugMatrix to calculate consistency as an external validation of the performance of AnimalGAN.

Results:

External Validation with DrugMatrix Data in Toxicity Assessment

An external validation of AnimalGAN was performed using a dataset derived from DrugMatrix[9]. It is a known that clinical pathology measurements can vary significantly between different experimental protocols or between different labs. For that reason, we analyzed the experiment data for the 70 common compounds (corresponding to 175 treatment conditions) tested in both TG-GATEs and DrugMatrix to establish a baseline concordance in their experiment settings. The overall average consistency between the two datasets for all 25 common measurements was 81.20%. For the external validation with 717 treatment conditions, the consistency between AnimalGAN generated results and the real data from DrugMatrix was 82.85%. **Figure 4a** shows a comparison of the consistency of the baseline settings for all the 25 measurements based on experiment data and the AnimalGAN results. Moreover, we compared the chemistry space of 110 training compounds against the 355 external validation compounds and found that they were not significantly overlapped (**Figure 4b**).

Discussion:

It is important to note that selecting a validation dataset for AnimalGAN is a challenge since different experimental study designs can lead to different test results. We established several criteria to select a reasonable validation set in our study: (1) same rat strain and sex, (2) a similar repeated dose study design as TG-GATEs, (3) common compounds tested by TG-GATEs to establish a baseline in comparison, and (4) contained clinical pathology measurements that significantly overlapped with those tested by TG-GATEs. DrugMatrix met all these criteria with several minor deviations. The first consideration is about the age of rats; DrugMatrix used rats of 5-7 weeks while TG-GATEs used rats that were 6-weeks, which might not be significant but worth to mention. The second consideration is about the testing strategy to determine Maximum Tolerated Dose (MTD); DrugMatrix applied a 5-day dose range finding method while TG-GATEs worked on a 7-day protocol, which might also not result in significant difference between two studies.

The third consideration is in defining the dose level, which might have a significant impact; DrugMatrix applied both MTD and therapeutic dose to define dose levels whereas TG-GATEs only relied on MTD. The fourth consideration regards control groups; DrugMatrix used a shared control while TG-GATEs applied time-matched controls. In addition, the source and purity of compounds might be different between two datasets, which we do not have information. Considering these differences between the two study designs, around 81% consistency in toxicity assessment for the common compounds using the experimental data seems reasonable. Consequently, the observed ~83% consistency of AnimalGAN results against the DrugMatrix experiment data demonstrates its potential application for toxicity assessment.

Figure 4. External Validation of AnimalGAN. **a** Comparison of Consistency in Toxicity Assessment Outcomes based on experiment data and AnimalGAN-generated data for all 25 measurements. The blue (left) bars represent the consistency in toxicity assessment outcomes between the experiment data from TG-GATEs and DrugMatrix. The orange (right) bars represent the consistency in toxicity assessment outcomes between the AnimalGAN-generated measurements and the experiment data tested in DrugMatrix. **b** Visualization of chemistry space of the 110 training compounds and the 355 external validation compounds. Each point depicts one compound.

2. Why are 1649 tx conditions? Is it $138 * 4 * 3$ (comp/duration/dose) = 1656? Please describe it in detail.

This is a reasonable question. Animals might be lost from study (died), particularly at high doses and long treatment durations. As a result, some compounds don't have the complete testing results as designed in

the TG-GATEs database. For example, there are no testing results for the compound “meloxicam” at high dose level treated for 28 days.

All treatment conditions used are listed in **Supplementary Table S3**.

3. Please summarize relevant GAN in M&M with references

There are three types of GAN designs that were integrated to construct AnimalGAN; these are Vanilla GAN, conditional GAN, and WGAN. In the M&M, each method was introduced, discussed and summarized with references. A similar comment was made by reviewer#2 who requested “a more detailed description of how the study conditions were encoded” (see comment 15 below). To address both comments, we have included a more detailed description in the Materials and Methods of how the treatment conditions were encoded. Specifically:

- Under “Generator Architecture” of M&M, we changed “All these four inputs were concatenated into a 3656-dimensional vector and used as input to pass into the generator G ,” to “All these four inputs were concatenated into a 3656-dimensional vector and scaled to [-1,1] as input to pass into the generator G ,”
- Under “Discriminator architecture” we have added “The 38-dimensional vector of clinical pathology measurements and 1828-dimensional vector of treatment conditions (i.e., descriptors + dose + time) were concatenated into an 1866-dimensional vector and scaled to [-1,1] as an input to the discriminator.”

4. Is there any missing in duration? How do you deal with imbalances of dose?

As far as we understand that, for each compound/dose/time condition, TG-GATEs data was from 5 rats. Some rats were lost from study, which mostly occur at high dose and/or long treatment duration. Overall, the missing data in terms of dose and time points are minimal, and does not raise any concern regarding imbalance in either dose or time points.

5. Please present a figure of architecture of generator and discriminator.

Thank you for the suggestion. In the M&M section, we described the architectures of the generator and the discriminator of the AnimalGAN. The architecture of the AnimalGAN is illustrated in **Figure 1a** and **Supplementary Table S5**. Additionally, we also include a detailed description of both the generator and the discriminator as **Supplementary Figure S4** in the revised version in a graphic representation as shown below:

Supplementary Figure S4. Graphic representation of the AnimalGAN. (A) Generator. (B) Discriminator.

6. Cross validate component splitting with k-folding and present Standard deviation.

Both cross-validation and hold-out validation are commonly used in internal validation. We chose the hold-out validation in our study to estimate the AnimalGAN performance. We specifically placed the emphasis on real-world application, as such that each hold-out validation represents different extreme conditions (i.e., the hold-out drugs vary significantly from those used in training, both in terms of chemical structure, drug class, and the year of FDA approval). We found that AnimalGAN can perform well in all these extreme hold-out validation situations. With that observation, we strongly believe that a k-fold cross-validation will not lead to a different observation, particularly since the prediction accuracies were so high (~98%) in these extreme hold-out scenarios. Moreover, in the revision, we also included an external validation with DrugMatrix dataset in which 355 compounds were not tested in TG-GATEs. Given the good performance on the external dataset, we feel that including a k-fold cross validation might not add too much value.

7. What's all these four inputs with 3656?

Inputs of generator, including 1826 descriptors for each compound, 1828-dim noise, 1-dim dose level and 1-dim treatment duration. We have now clarified this information and their processing in the revised version as indicated below:

The generator G received four inputs, with the first three inputs as the treatment condition $c = \text{concat}(s, d, t)$. The first one was the molecular representation of a chemical (s), an 1826-dimensional vector of molecular representation by using Mordred. The second input was the dose level of a compound administrated to the rat (d). For the animal studies in the Open TG-GATEs database, the ratio among low, middle, and high dose levels was set as 1:3:10 with the high dose equal to the Maximum Tolerance Dose determined in a 7-day dose finding study, with the same setting applied here. The third input denotes the time point (3, 7, 14, and 28 days) a rat (t) was treated. The final input was an 1828-dimensional noise vector (n) sampled from a normal distribution. All these four inputs were concatenated into a 3656-dimensional vector and scaled to $[-1, 1]$ as input to pass into the generator G , resulting in $\tilde{x} = G(c, z)$, where \tilde{x} was a generated vector of clinical pathology measurements.

8. There is a lack of parameter and model searching, and ablation studies.

We apologize for not clearly presenting our modeling process involving parameter and model search and ablation studies. In the revised version of the **Supplementary Information**, we have added a section to explain this process.

Hyper-Parameter Tuning

To determine whether the model has achieved statistical convergence, we evaluate the cumulative loss of AnimalGAN across 1000 epochs. It's important to note that not all generated records are biologically valid. Hence, we calculate the ratio of valid records among all generated ones, referring to this as the "valid records ratio". This ratio serves as one criterion to demonstrate the model's biological relevance and performance. For the generated testing results, only those that pass the blood count check are used for the following analysis, and the invalid records are discarded.

Learning rate

We utilized the Adam optimizer to optimize the loss function, employing three key parameters: the initial learning rate, the decay rate of the first-order momentum of the gradient (denoted as beta1 or b1), and the decay rate of the second-order momentum of the gradient (denoted as beta2 or b2). Since

our training dataset is relatively small, we used correspondingly small initial learning rates, b1 values and b2 values. In this step, we fixed other parameters to find a more suitable learning rate, b1 and b2 values. Ultimately, we selected an initial learning rate of $1e-7$, a beta1 value of 0.8, and a beta2 value of 0.95. These values were then fixed for subsequent rounds of hyper-parameter tuning.

Supplementary Figure S7. Comparison for different learning rates when other parameters are fixed.

Batch size

We also conducted a search for an appropriate batch size. As a result of this search, a batch size of 128 was chosen for subsequent tuning steps.

Supplementary Figure S8. Comparison for different batch sizes when other parameters are fixed.

Number of discriminator Iterations

In each iteration, we trained the generator once and the discriminator multiple times. We evaluated the influence of varying the number of discriminator iterations, considering values of 5, 10 and 20. Ultimately, we determined that using 5 iterations for the discriminator yield optimal results.

Supplementary Figure S9. Comparison for different numbers of discriminator iterations when other parameters are fixed.

Noise length

A random noise is an input to the generator and the generator maps the noise distribution to the distribution of clinical pathology measurements. We also evaluated the impact of the noise length (the dimension of the noise vector). Specifically, we tested noise lengths of 16, 38 (equal to the number of clinical pathology measurements), 64, 128 and 1828 (the dimension of the treatment conditions: descriptors + time + dose). **Supplementary Figure S10** indicated that the convergence and valid records ratio do not vary much across the various noise lengths. As a result, we opted to retain a noise length of 1828 for our model.

Supplementary Figure S10. Comparison for different noise length when other parameters are fixed.

9. There should be Bland Altman analysis on 38 measurements, which need to be evaluated in viewpoint of systemic bias.

Thank you for the suggestion. We did an analysis for all 38 measurements across all the treatment conditions, which was included in a **Supplementary Figure S12**. We did not observe a consistent bias of AnimalGAN over experimental results.

Supplementary Figure S12. Bland Altman analysis on 38 measurements.

10. Fig 2 should be tabulated with statistical comparison.

Thank you for the suggestion. We have listed the performance of AnimalGAN and 12 QSAR models in the **Supplementary Table S1**. Additionally, we have transformed **Figure 2** into a violin plot for a better data visualization.

Figure 2. Comparisons of AnimalGAN results with QSAR predictions for the test set of all 38 clinical pathology measurements. For each measurement, the performance of the 12 QSARs was represented in a violin plot while the performance of AnimalGAN was denoted by a golden star. The plot was z-score scaled for an improved visual inspection. AnimalGAN exhibited consistently smaller MSE than what can be achieved with QSARs.

11. Reviewer #2 (Remarks to the Author):

In their article titled "AnimalGAN: A GAN Model for Clinical Pathology Assessment," Chen et al. propose the use of Generative Adversarial Networks (GANs) as an alternative to animal studies in evaluating chemical and drug safety. Their model, AnimalGAN, successfully simulated 38 rat clinical pathology measures and showed comparable results to real animal data in hepatotoxicity assessment, surpassing conventional approaches. AnimalGAN's virtual experiment accurately ranked the hepatotoxicity of three structurally similar drugs, demonstrating its potential for reducing and refining animal use in research.

The paper is well-written and easily understandable, with high-quality display items that effectively enhance the clarity of the information presented.

We thank the reviewer for their compliments on the paper, which we appreciate very much!

12. How does the evaluation of the model's performance on the test set demonstrate its effectiveness and practicality for real studies?

This is an excellent question; a similar question was also raised by Reviewer #1 (#1 question). As a result, an extensive analysis was included in the revised paper to demonstrate the performance of AnimalGAN through an external validation on 355 compounds that were not seen by the TG-GATEs dataset. We detailed in the revision the external validation process in M&M, Results and Discussion. All new texts in these three sections are presented in response to Reviewer 1, question 1.

13. In addition to reporting the mean squared error (MSE), I am interested in understanding how the model performs under an ablation study.

Thank you for this question. In this revision, we included an ablation study in the **Supplementary Information**.

Using 2D molecular descriptors only to build model

In the main manuscript, we report the AnimalGAN, which utilizes both 2D and 3D molecular descriptors generated by Mordred. In addition, we also conducted a study using 2D molecular descriptors only. Similar to the reported AnimalGAN, we conducted parameter searches and model training, and the final model was employed to generate clinical pathology measurements for the same test dataset used in the reported AnimalGAN. The results from the model based on 2D descriptors demonstrated comparable performance to that of the AnimalGAN and the figure below is included as **Supplementary Figure S11** in the revised **Supplementary Information**.

Supplementary Figure S11. Boxplot of RMSE - Root Mean Square Error and Cosine Similarity between generated synthetic data and real animal testing data for treatment conditions in the test set.

14. Furthermore, I would suggest interpolating the molecular descriptor between DSMO* and known drug-induced liver injury (DILI) compounds to observe how the clinical pathology findings change. It would be valuable to keep the noise pattern schedule fixed for the intermediate state during this analysis.

*: DMSO

Indeed, DMSO is a commonly used vehicle, not only for DILI. To the best of our knowledge, we have not seen any literature reports about the interplay between DMSO and DILI compounds in terms of DILI severity. Therefore, even including DMSO in the modeling process, we might have difficulties to interpret the modeling results. With that said, this is an interesting question and we may look into it in the future.

15. The authors should provide a more detailed description of how the study conditions were encoded and how the clinical pathology findings were normalized across different conditions to correct for plain batch effects.

Thank you for this question. We have included a more detailed description of how the treatment conditions were encoded in the Materials and Methods section of our revised manuscript. Specifically,

- Under “Generator Architecture” of M&M, we changed “All these four inputs were concatenated into a 3656-dimensional vector and used as input to pass into the generator G ,” to “All these four inputs were concatenated into a 3656-dimensional vector and scaled to [-1,1] as input to pass into the generator G ,”
- Under “Discriminator architecture” of M&M, in the beginning we add “The 38-dimensional vector of clinical pathology measurements and 1828-dimensional vector of treatment conditions (i.e., descriptors + dose + time) were concatenated into an 1866-dimensional vector and scaled to [-1,1] as an input to the discriminator.”

Since the TG-GATEs data were generated in accordance with a standard protocol, we did not perform normalization to the clinical pathology measurements across different conditions.

16. Regarding the evaluation, I would like to see the mean squared error (MSE) plotted over i) time points (3,7,14 and 28) after treatment and ii) dose level to ensure that the model is not learning trivialities that primarily minimize the overall MSE by focusing on the time or dose component.

Thank you for this suggestion. We have provided the MSE plotted over time points and dose levels in the **Supplementary Figure S13**. By examining the MSE trends over time points and dose levels, our model is learning meaningful patterns and not simply minimizing the overall MSE by focusing on specific components. The newly added section in the revised **Supplementary Information** is as follows:

Evaluation based on MSEs across dose levels and time points

To comprehensively assess the performance of the AnimalGAN, we employed the mean squared error (MSE) as measure to compare the AnimalGAN-generated results against real experiment data for the treatment conditions in the test set. specifically, we focused on the following aspects:

Time Points:

We conducted an evaluation of the MSE over different time points after treatment (3, 7, 14, and 28 days), that is scarified period of 4, 8, 15 and 29 days. This analysis helps us ascertain that the model's

performance is not driven by trivial optimizations that predominantly minimize the overall MSE by emphasizing specific time intervals. **Supplementary Figure S13 (A)** illustrates the MSE values across these time points.

Dose Level:

Additionally, we evaluated the MSE across various dose levels. This examination ensures that the model's performance is not skewed towards minimizing MSE by focusing solely on certain dosage levels. The corresponding **Supplementary Figure S13 (B)** provides a visualization of MSEs across different dose levels.

The depiction of MSE trends over time points and dose levels serves as compelling evidence that our model captures meaningful patterns and avoids overemphasizing specific components solely for the purpose of minimizing MSE.

Supplementary Figure S13. MSEs over (A) time points and (B) dose levels.

17. The choice of input representation appears somewhat arbitrary. Have other molecular descriptors been tested as well? Additionally, what impact does the size of the 1828-dimensional noise vector have on the model's performance?

Thank you for this question. We didn't apply the molecular descriptors generated with different software packages. Having said that, when we removed 213 3D-descriptors from the modeling, we were able to generate a comparable model (please see the detailed answers for Question #13).

Regarding the impact of the size of the 1828-dimensional noise vector, this is a good question. In response, we have integrated an assessment of the impact of the noise vector's length into the revised version of the **Supplementary Information**. Our evaluation employs two criteria, statistical convergence and "valid records ratio", to assess model's performance and biological relevance. The relevant parts are provided below:

To determine whether the model has achieved statistical convergence, we evaluate the cumulative loss of AnimalGAN across 1000 epochs. It's important to note that not all generated records are biologically valid. Hence, we calculate the ratio of valid records among all generated ones, referring to this as the "valid records ratio". This ratio serves as one criterion to demonstrate the model's biological relevance and performance. For the generated testing results, only those that pass the blood count check are used for the following analysis, and the invalid records are discarded.

Noise length

A random noise is an input to the generator and the generator maps the noise distribution to the distribution of clinical pathology measurements. We evaluated the impact of the noise length (the dimension of the noise vector). Specifically, we tested noise lengths of 16, 38 (equal to the number of clinical pathology measurements), 64, 128 and 1828 (the dimension of the treatment conditions: descriptors + time + dose). **Supplementary Figure S10** indicated that the convergence and valid records ratio do not vary much across the various noise lengths. As a result, we opted to retain a noise length of 1828 for our model.

Supplementary Figure S10. Comparison for different noise length when other parameters are fixed.

I see great potential in this work, and the chosen topic is highly relevant. However, I believe that there is room for improvement in the manuscript as it stands. Therefore, I request a revision to address the identified areas for improvement.

Reviewer #3 (Remarks to the Author):

18. The article presents the development of a new approach methodology (NAMs) using AI to simulate rats and predict 38 clinical pathology measurements, including 21 hematologic and 17 clinical chemistry parameters. As a training set, the authors used a subset of in vivo experiments available on the Open TG-GATEs platform. The dataset used for training included 8,078 rats treated with 138 compounds under 1,649 treatments. The model was validated by comparing it with QSAR models developed for the same purpose. Finally, a case study was conducted for three thiazolidinediones, and a population of 100,000 rats was generated.

The approach developed here is novel and leverages modern AI techniques to address toxicology problems. The paper is well-written, and the authors have made an effort to describe the methods in the journal format. Unfortunately, my review is limited to the content of the publication, as the links to the code and data (<https://github.com/XC-NCTR/AnimalGAN> , <https://dbarchive.biosciencedbc.jp/en/open-tggates/download.html>) are not working or are empty. The authors should provide the following information for a more detailed review:

The methods for developing the AI model involved grid optimization, cosine similarity, and RMSE scoring. However, the authors did not discuss the issue of overfitting in the model. I was surprised to see a performance gap between epoch 5000 and epoch 6000 during the training phase, as well as almost perfect prediction error plots. The authors should address the issue of overfitting and attempt to estimate its impact.

We understand the reviewer's question and appreciate the concern re the possible "overfitting" in our training process. This is a common concern in developing a predictive model through training processes, such as developing a QSAR model or other types of predictive models using conventional ML methods (e.g., RF and SVM). This overfitting issue is normally evaluated through internal validation, followed by an external validation. We took this route to mitigate "overfitting" in AnimalGAN. Specifically, we selected the model at 6000 epochs, followed with hold-out internal validations and an external validation on the DrugMatrix dataset. In our study, convergence in AnimalGAN was determined by the minimum loss in the loss function as generally practiced in this field. Indeed, we observed that the convergence was obtained at 1000 epochs, raising concerns as to whether this could be "overfitting". Therefore, we generated additional models at every 1000 epochs onwards. For each model, three performance metrics (i.e., RMSE,

cosine similarity and t-SNE) were examined and compared. As shown in **Supplementary Figure S5**, the model quality reached a peak at 6000 epochs and then decreased at 7000 epochs. Thus, the model from 6000 epochs was selected as the AnimalGAN model, which was subsequently evaluated with the 28 hold-out compounds (i.e., 332 treatment conditions) and evaluated with 355 compounds from DrugMatrix, both “28” and “355” compounds were not presented in the training model.

Supplementary Figure S5. Evolution of model performance during training. Distributions of (A) cosine similarities and (B) RMSEs between generated data and real animal testing data for all the treatment conditions in training set along with training. (C) t-SNE visualization of generated data and real data for treatment conditions in the training set at different training epochs. Each point depicted one treatment condition.

19. Another concern I have is regarding the comparison with QSAR models. The supporting information document shows a list of chemicals with similarity scores, and almost all of them have scores above 0.9. If this is the case for the 138 compounds used in the training, it suggests that QSAR models built from molecular descriptors may not perform well due to the limited diversity of the chemicals. The authors should take this into consideration and discuss the limitations of their validation.

This is an excellent point. In the first paragraph of “Discussion”, we change “By comparing to 12 conventional QSAR methods for each of 38 clinical pathology measures using the exact same input, AnimalGAN outperformed QSARs in prediction of all the clinical pathology measurements.” To “By comparing to 12 conventional QSAR methods for each of 38 clinical pathology measures using the exact

same input, AnimalGAN outperformed QSARs in prediction of all the clinical pathology measurements within the chemistry space pertaining to TG-GATEs.”

20. Furthermore, the authors should discuss the applicability of this model. How far can the model be extended with only 138 chemicals considered as input, and what considerations should future users have when utilizing this model? The issue of chemical diversity should also be addressed. If I understand the table in the supporting information correctly, it indicates poor diversity.

This is another excellent question. In the revised version, we applied AnimalGAN to 355 compounds from DrugMatrix (text relating to this external validation is in response to Question #1 of Reviewer #1). These 355 compounds from DrugMatrix that are not by TG-GATEs are in the chemistry space different from the training set (**Figure 4**). However, we were still able to achieve a very good result in comparison to the experimental results. Although the applicability domain was not explicitly analyzed in this study, it seems the model can be extrapolated quite far.

21. Minor comments:

- Could the authors provide more details on how they calculated the background control?

Thank you for this question. We added more details on how the background control distribution was estimated in the revised version. The added text is provided below.

Specifically, in order to construct a background control distribution, we calculated the Cosine Similarities and RMSEs between every pair of treatment conditions (i.e., compound/time/dose combination) based on their lab-tested clinical pathology profiles. We then compared the distributions of Cosine Similarities and RMSEs between the profiles generated by AnimalGAN and the corresponding profiles obtained from TG-GATEs against the background control distributions. We employed Wilcoxon rank-sum tests to compare these distributions against the background control distributions.

22. - Please check the author list, as there are differences between the manuscript and the supporting information.

Thank you for bringing this to our attention. We apologize for the discrepancy in the author list. We have corrected the author list in the supporting information.

23. - In Figure S6, please add the percentage of variance explained for each principal component in the PCA. Also, comment on the poor overlap between the training and test sets. Were the test samples randomly selected or optimized?

Thank you for this suggestion. We have added the percentage of variance explained for each principal component in Figure S6.

Supplementary Figure S6. Visualization of structural similarities of all the 138 compounds. The pairwise structural similarities between any two of the 138 compounds were calculated based on their Mordred molecular representations. Each point depicted one compound.

Regarding the test sample selection process for the structure-based splitting, we retained compounds whose chemical structures were far different from those that were used to develop AnimalGAN model in the test set.

24. - I found it difficult to read and compare the performances in Figure 2. Using a non-circular figure would make it easier to read. Additionally, the square representing the performance of AnimalGAN could be improved.

Thank you for the suggestion. We have listed the performance of AnimalGAN and 12 QSAR models in the **Supplementary Table S1**. Additionally, we have transformed **Figure 2** into a violin plot for a better data visualization.

Figure 2. Comparisons of AnimalGAN results with QSAR predictions for the test set of all 38 clinical pathology measurements. For each measurement, the performance of the 12 QSARs was represented in a violin plot while the performance of AnimalGAN was denoted by a golden star. The plot was z-score scaled for an improved visual inspection. AnimalGAN exhibited consistently smaller MSE than what can be achieved with QSARs.

REVIEWERS' COMMENTS

Reviewer #1 (Remarks to the Author):

I suspect that the authors respond my questions and comments sufficiently. I have no further comments.

Reviewer #2 (Remarks to the Author):

The authors answered all my questions, I have no further comments and suggest to accept the manuscript in its current form.

Reviewer #3 (Remarks to the Author):

Dear Authors,

I thank you for your efforts in developing your manuscript. I appreciate the enormous work you've done to enhance the methods section and propose additional model validation. Most of my comments have been discussed, but I believe that one of my previous comments needs more attention:

From my previous comment:

Another concern I have is regarding the comparison with QSAR models. The supporting information document shows a list of chemicals with similarity scores, and almost all of them have scores above 0.9. If this is the case for the 138 compounds used in the training, it suggests that QSAR models built from molecular descriptors may not perform well due to the limited diversity of the chemicals. The authors should take this into consideration and discuss the limitations of their validation.

The authors responded:

This is an excellent point. In the first paragraph of "Discussion", we change "By comparing to 12 conventional QSAR methods for each of 38 clinical pathology measures using the exact same input, AnimalGAN outperformed QSARs in prediction of all the clinical pathology measurements." To "By comparing to 12 conventional QSAR methods for each of 38 clinical pathology measures using the exact same input, AnimalGAN outperformed QSARs in prediction of all the clinical pathology measurements within the chemistry space pertaining to TG-GATEs."

The authors did not elaborate on the similarity score and the diversity of the chemical set. Please provide further details in your response. Additionally, it is important to note that the model has been developed within a relatively limited chemical space, which may affect the applicability of this approach to any other types of chemicals, such as environmental chemicals.

REVIEWERS' COMMENTS

Reviewer #3 (Remarks to the Author):

Dear Authors,

I thank you for your efforts in developing your manuscript. I appreciate the enormous work you've done to enhance the methods section and propose additional model validation. Most of my comments have been discussed, but I believe that one of my previous comments needs more attention:

From my previous comment:

Another concern I have is regarding the comparison with QSAR models. The supporting information document shows a list of chemicals with similarity scores, and almost all of them have scores above 0.9. If this is the case for the 138 compounds used in the training, it suggests that QSAR models built from molecular descriptors may not perform well due to the limited diversity of the chemicals. The authors should take this into consideration and discuss the limitations of their validation.

The authors responded:

This is an excellent point. In the first paragraph of "Discussion", we change "By comparing to 12 conventional QSAR methods for each of 38 clinical pathology measures using the exact same input, AnimalGAN outperformed QSARs in prediction of all the clinical pathology measurements." To "By comparing to 12 conventional QSAR methods for each of 38 clinical pathology measures using the exact same input, AnimalGAN outperformed QSARs in prediction of all the clinical pathology measurements within the chemistry space pertaining to TG-GATEs."

The authors did not elaborate on the similarity score and the diversity of the chemical set. Please provide further details in your response. Additionally, it is important to note that the model has been developed within a relatively limited chemical space, which may affect the applicability of this approach to any other types of chemicals, such as environmental chemicals.

Dear Reviewer,

Thank you for your valuable feedback and for recognizing the efforts to enhance our manuscript. We appreciate your continued engagement with our work.

Regarding your specific concern about the comparison with QSAR models and the diversity of the chemical set, we agree that this is an important aspect to discuss.

To clarify, the similarity scores presented in the supplementary information represent relative scores used for structure-based splitting, not absolute similarities among compounds. However, due to the small dataset we used, we acknowledge the limited diversity of the chemical set. We have incorporated this acknowledgement in the revised "Discussion" section as follows:

"We acknowledge that AnimalGAN was developed within a relatively restricted chemical space, which may impact the generalizability of our findings. The conclusion that AnimalGAN outperforms conventional QSARs is based on the dataset we used, and caution should be exercised when extending our results to diverse chemical classes or environmental compounds."